# TRAIN MONOLINGUAL, INFER BILINGUAL

**Alaeddin Selçuk Gürel**[*]
Huawei Turkey R&D Center
alaeddin.selcuk.gurel2@huawei.com

**Aydın Gerek**[*]
Huawei Turkey R&D Center
aydin.gerek@huawei.com

## ABSTRACT

Cross-lingual transfer learning has been studied at depth. While many methods have been developed for pretraining or fine-tuning on monolingual, multilingual and parallel corpora with the purpose of predicting on a low-resource monolingual test set; in this paper we investigate the feasibility of training a text classifier on a monolingual training set and predicting on a parallel test set, jointly utilizing both languages at inference time only.

## 1 INTRODUCTION

Cross-lingual transfer learning, especially in the context of training in one language and testing in another language has been studied thoroughly (Pikuliak et al., 2021) (Hangya et al., 2018) (Antony et al., 2020) (Bel et al., 2003). In this paper using a unique dataset we study the problem of testing on parallel data. Specifically, we fine-tune a multilingual BERT model on English language paper titles from ArXiv metadata and then run inference on another dataset composed of thesis titles written in both Turkish and English. To the best of the author's knowledge, this is a novel task. We share the data and the baseline of this new task[1]. We compare the model's success in both languages (as has been done in many papers), but more importantly, show that it is possible to increase model performance by predicting in both languages simultaneously.

## 2 DATASETS

We used two different binary classification datasets in this paper. Training data includes academic paper titles in English which are extracted from ArXiv [2]. with the target determined by whether the paper in question belongs to the CS.CL domain. The negative class was undersampled to match the positive class, resulting in a balanced dataset of 64k titles. The dataset was divided into three balance subsets which is 64% for training, 20% for testing, and 16% for the validation set.

The second dataset (which we will call the YOK dataset) includes academic paper articles scraped from the website of Turkey's Council of Higher Education [3], consisting of the titles of 200 masters/Ph.D thesis titles published in computer science departments across Turkey. There are 15 titles labeled as NLP, while the remaining 185 titles categorized as non-NLP. It has been manually labeled with the binary target of whether the thesis topic is in the NLP domain or not. There are two title fields in the YOK dataset; each sample comes with title of the thesis in English and Turkish.

## 3 METHODOLOGY

First, we fine-tune multilingual BERT [4], on the ArXiv dataset. Next, for comparison, we separately predict the Turkish titles and the English titles in the YOK dataset. Finally, we run joint prediction on both the Turkish and English titles simultaneously by averaging the logit outputs of the model.

---

[1] https://github.com/alaeddingurel/train_monolingual_infer_bilingual.
[*] These authors contributed equally to this work
[2] https://www.kaggle.com/datasets/Cornell-University/arxiv
[3] https://tez.yok.gov.tr/
[4] https://huggingface.co/bert-base-multilingual-cased

We also tried out a more general weighted average with hyperparameter $t$.

$$\overrightarrow{joint\_logit} = t \, \overrightarrow{Turkish\_logit} + (1-t) \, \overrightarrow{English\_logit} \quad ; 0 < t < 1$$

## 4  EXPERIMENTAL RESULTS

The evaluation scores for each class for the Arxiv set can be seen in Table 1.

As can be seen in Table 2, precision and f1 scores are significantly improved by averaging logits, surpassing not just the Turkish-only scores but also the English-only scores.

Table 1: Evaluation Metrics for ArXiv test set

| Class | precision | recall | f1-score |
|-------|-----------|--------|----------|
| 0 | 96.19 | 95.29 | 95.74 |
| 1 | 95.33 | 96.22 | 95.77 |

Table 2: Evaluation Metrics for paper titles from YOK

| Language | precision | recall | f1-score | accuracy |
|----------|-----------|--------|----------|----------|
| English | 32.25 | **86.66** | 47.27 | 85.50 |
| Turkish | 27.90 | 80.00 | 41.37 | 83.00 |
| Mean (t=0.5) | 34.21 | **86.66** | 49.05 | 86.5 |
| W. Mean (t=0.3) | **35.13** | **86.66** | **50.00** | **87.00** |

## 5  CONCLUSION AND FUTURE WORK

There are many situations where it is not possible or desirable to train a model from scratch or even fine-tune it. This is especially the case where the model is large and compute resources are scarce. In this paper, we've shown a simple inference trick that can be applied when the test data is bilingual, which increases model performance significantly. While the natural application opportunities for this technique will be scarce since most naturally occurring data is not bilingual, In future work, it should be possible to adapt this technique to low-resource monolingual data by use of machine translation systems.

Other potential improvements to this work include leveraging statistical machine translation era word alignment systems for token labeling tasks. Averaging over three more languages, especially with the help of machine translation systems would also likely yield further improvements. It would be also interesting to investigate whether averaging logit outputs of a linear classifier applied to sentence representations from earlier (as opposed to final) layers of the BERT architecture also similarly improve performance.

### URM STATEMENT

Both authors were born in and currently are located in Turkey. As such they fulfill the geographical partion of the URM criteria. The first author is also younger than 30 years old.

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
