# OpenReview forum: "Train Monolingual, Infer Bilingual"
_ICLR.cc/2023/TinyPapers — Submitted to Tiny Papers @ ICLR 2023_

### Official Review · Reviewer_4av8 · 2023-03-30

**Confidence:** 4

**Summary Of Contributions:**

This paper constructs a new binary text classification task: determine whether an academic paper belongs to the CS.CL domain based on its title. The authors use English dataset to fine-tune a multilingual BERT, and then compare the model's inference performance for English, Turkish and the joint prediction by averaging the logit for both languages. The experimental results show that averaging logit of both languages is better than inferencing on source or target language alone.

**Rating:**

Great Start (GS): a submission which meets some of the reviewing criteria but has room for improvement

**Strengths And Weaknesses:**

Strengths:

1. Build datasets for a new text classification task.

2. Analyze how the method can be used in real world (using machine translation system), and list a number of perspectives for further research.

Weaknesses:

1. Since only one target language is included in this paper, it is worth further verifying the effectiveness of the method in more languages.

2. The experimental section does not analyze why averaging logit improves performance. As can be seen from the tables, the model performance difference on domains (ArXiv vs YOK) is significantly higher than the performance difference on languages (English vs Turkish on YOK). Having a cross-domain problem nested within a cross-lingual problem may be one of the reasons why averaging logit improves performance. Specifically, because the performance difference between Yok-English and Yok-Turkish is relatively small (perhaps thanks to mBERT's good multilingual ability), ensemble their logit could improve predictive performance. However, in the case of in-domain, such as translating the ArXiv test set into Turkish, the performance difference between ArXiv-English and ArXiv-Turkish would be quite large, and it is likely that averaging logit is worse than the result of inferencing only on English.

3. There are no statistics on the number of positive and negative examples of the datasets, making the results in the table unintuitive.


**Suggested Changes:**

1. Include a wider variety of target languages.

2. Analyze the experimental results.

3. Contain brief statistics on the datasets.

---

### Official Review · Reviewer_89bF · 2023-03-30

**Confidence:** 4

**Summary Of Contributions:**

The authors aim to improve text classifier performance for a low resource language such as Turkish. They investigate the effects of fine-tuning a multilingual model on monolingual training corpora and predicting on a multilingual parallel set.

**Rating:**

Great Start (GS): a submission which meets some of the reviewing criteria but has room for improvement

**Strengths And Weaknesses:**

*Strengths*
- The authors present an interesting methodology to tackle the problem of text classification in low resource languages.
- Clarity: the findings are communicated effectively.
- Reproducibility: the authors will share their data and code.

*Weaknesses*

The goal of the paper should be framed differently, or the methodology presented should be re-considered. The aim of the study is clear from the beginning, but the methodology followed is not the best for quantifying model performance in classification of Turkish titles when training in English. It is hard to interpret results, as the drop in performance for Turkish is highly due to the change of task domain (you can see that as performance differences between English and Turkish are not that big) and not due to test in a new language. Since the performance for both languages is already low, the gain from applying the weighted average of their logits is relatively small.

As the meaning of the positive class is not the same in arxiv and YOK datasets, it would be beneficial to report more statistics on the content of each dataset split. Like, how many titles in the training data belong to NLP?

**Suggested Changes:**

- Read what I wrote in Weaknesses. If you do not want to modify the set up, I would recommend to rephrase the framing of your work.
- The second sentence in the abstract needs to be rephrased, the verb is missing is the second clause (after the comma).
- In general, revise the paper for typos (e.g., capital letter after punctuation marks, Table should be capitalized)
- The proportions of train/test splits should be included.
- Table 1 should be mentioned in the text.
- The tables report precision, recall and F1 scores. Usually, unlike otherwise stated by the authors, it is sufficient to report F1 score because it reflects the harmonic mean of precision and recall. In other words, it is redundant to report all 3 metrics (unless it helps to draw a conclusion that is 'hidden' in the F1 score, but here it is not the case).
- Analyse and discuss your results: The proportion of examples in the test split would help to understand the results. The big gaps between metrics presented in Table 2 should be explained. Also, the fact that recall is the exact same value for 3 different scenarios is surprising and worth revisiting/explaining.

---

### Author Response · Authors · 2023-05-31
**Thank you for reviews**

We thank the area chair and reviewers for their valuable comments. We made some revisions. We wish to opt-in for archival.

---

### Meta-Review · Area_Chair_dHi7 · 2023-04-06

**Recommendation:** Invite to archive
**Confidence:** 4

**Metareview:**

Binary classification task used to show model performance for a low resource language, where they investigate the effects of fine-tuning a multilingual model on monolingual training corpora and predicting on a multilingual parallel set.

The research area is vital. The aim of the research is communicated well. However, the methodology needs to be clarified. Proofread the paper carefully. Ensure Tables are mentioned in the text; need clarity on the experimental setup. Also, where possible, explain the findings.

More importantly, the details of the dataset will add value to the overall clarity of the work.


**Summary:**

Binary classification task used to show model performance for a low resource language, where they investigate the effects of fine-tuning a multilingual model on monolingual training corpora and predicting on a multilingual parallel set.

**Reason For Not Giving A Higher Recommendation:**

Needs improvement in clarity of research methodology, dataset and findings.

**Reason For Not Giving A Lower Recommendation:**

N/A

---

### Decision · Program_Chairs · 2023-04-08

Invite to archive